# Neuroinflammatory Reactions in the Brain of 1,2-DCE-Intoxicated Mice during Brain Edema

**DOI:** 10.3390/cells8090987

**Published:** 2019-08-27

**Authors:** Xiaoxia Jin, Tong Wang, Yingjun Liao, Jingjing Guo, Gaoyang Wang, Fenghong Zhao, Yaping Jin

**Affiliations:** 1Department of Occupational and Environmental Health, School of Public Health, China Medical University, Shenyang 110122, Liaoning, China; 2Department of Occupational and Environmental Health, School of Public Health, Shenyang Medical College, Shenyang 110034, Liaoning Province, China; 3Department of Physiology, China Medical University, Shenyang 110122, Liaoning, China

**Keywords:** 1,2-dichloroethane intoxication, nuclear factor-κB, neuroinflammatory reactions, proinflammatory factors, matrix metalloproteinases-9, brain edema

## Abstract

We previously reported that expression of matrix metalloproteinase-9 (MMP-9) mRNA and protein was upregulated during 1,2-dichloroethane (1,2-DCE) induced brain edema in mice. We also found that the p38 mitogen-activated protein kinase (p38 MAPK) signaling pathway resulted in MMP-9 overexpression and nuclear factor-κB (NF-κB) activation in mice treated with 1,2-DCE. In this study, we further hypothesized that inflammatory reactions mediated by the p38 MAPK/ NF-κB signaling pathway might be involved in MMP-9 overexpression, blood–brain barrier (BBB) disruption and edema formation in the brain of 1,2-DCE-intoxicated mice. Our results revealed that subacute poisoning by 1,2-DCE upregulates protein levels of glial fibrillary acidic protein (GFAP), ionized calcium-binding adapter molecule 1 (Iba-1), interleukin-1β (IL-1β), vascular cell adhesion molecule-1 (VCAM-1), intercellular adhesion molecule-1 (ICAM-1), inducible nitric oxide synthase (iNOS) and p-p65 in mouse brains. Pretreatment with an inhibitor against p38 MAPK attenuates these changes. Moreover, pretreatment with an inhibitor against NF-κB attenuates alterations in brain water content, pathological indications notable in brain edema, as well as mRNA and protein expression on levels of MMP-9, VCAM-1, ICAM-1, iNOS, and IL-1β, tight junction proteins (TJs), GFAP and Iba-1 in the brain of 1,2-DCE-intoxicated mice. Furthermore, pretreatment with an inhibitor against MMP-9 obstructs the decrease of TJs in the brain of 1,2-DCE-intoxicated mice. Lastly, pretreatment with an antagonist against the IL-1β receptor also attenuates changes in protein levels of p-p38 MAPK, p-p65, p-IκB, VCAM -1, ICAM-1, IL-1β, and Iba-1 in the brain of 1,2-DCE-intoxicated-mice. Taken together, findings from the current study indicate that the p38 MAPK/ NF-κB signaling pathway might be involved in the activation of glial cells, and the overproduction of proinflammatory factors, which might induce inflammatory reactions in the brain of 1,2-DCE-intoxicated mice that leads to brain edema.

## 1. Introduction

Synthetic halohydrocarbon 1,2-dichloroethane (1,2-DCE) is mainly used in the manufacture of polyvinyl chloride worldwide, and also as an industrial solvent both in China and other countries. Thus, industrial workers may be exposed to high concentrations of 1,2-DCE through the air [1,2]. Over the past few decades, accumulated evidence demonstrated that toxic encephalopathy is a serious, common disorder in workers suffering from subacute poisoning of 1,2-DCE, where brain edema is the main pathological consequence [3,4,5]. Nevertheless, little is known about the molecular mechanisms of 1,2-DCE-induced brain edema.

In previous studies, we found that matrix metalloproteinase-9 (MMP-9) was upregulated transcriptionally during the formation of brain edema that was induced by subacute poisoning of 1,2-DCE in mice. We also found that the p38 mitogen-activated protein kinase (p38 MAPK) signaling pathway was activated and involved in the overexpression of MMP-9 [6,7]. Studies thus far have demonstrated that MMP-9 is substantially overexpressed in many brain diseases, which acts as not only a proteolytic enzyme involved in the disruption of the blood–brain barrier (BBB), but also an inflammatory mediator playing an important role in neuroinflammation [8,9,10]. It is well-known that inflammatory reactions are involved in the pathogenesis of brain diseases, and that p38 MAPK act as a critical mediator in these reactions [11,12]. In addition, nuclear factor-κB (NF-κB) as a key transcription factor that also plays an important role in the inflammatory reaction by promoting the production of proinflammatory factors and inflammatory mediators [13,14].

Growing evidence has demonstrated that interleukin-1β (IL-1β) is the most important proinflammatory factor responsible for the innate immune response [9,15]. Some studies have further demonstrated that IL-1β could promote MMP-9 expression by activating NF-κB [16,17], and in turn, activation of NF-κB also enhances the production of IL-1β [18,19]. The positive feedback loop between production of IL-1β and activation of the NF-κB signaling pathway may serve to amplify inflammatory signals and exacerbate brain injury. Moreover, it has been reported that inflammatory mediators, including intercellular adhesion molecule-1 (ICAM-1), vascular cell adhesion molecule-1 (VCAM-1) and inducible nitric oxide synthase (iNOS), are overexpressed through activation of the NF-κB signaling pathway and are believed to contribute to the brain edema formation [20,21]. Furthermore, evidence indicates that neuroglial cells may be activated in response to exogenous stimuli, resulting in the production of proinflammatory factors that are released by activated microglia and astrocytes, leading to neuroinflammation [22,23,24].

Although we have found that NF-κB could be activated through p38 MAPK signaling pathway during the course of 1,2-DCE-induced brain edema in mice [6], it is still unknown whether activation of NF-κB could enhance MMP-9 expression and whether BBB integrity might be disrupted by excessive MMP-9. Thus, in the present study, we further hypothesized that the p38 MAPK/NF-κB signaling pathway might be involved in the activation of glial cells and as a consequence, the production of the proinflammatory factor IL-1β and the inflammatory mediators that induce neuroinflammation lead to BBB disruption and brain edema formation in 1,2-DCE-intoxicated mice. As far as we know, this is the first study investigating the involvement of neuroinflammation in 1,2-DCE-induced brain edema in mice.

## 2. Materials and Methods

### 2.1. Animal Care and Use Statement

The present experiments were performed after approval by the Scientific Research Committee of China Medical University and conformed to the Guide for the Care and Use of Laboratory Animals from National Institutes of Health. The project identification code was IACUC: NO.2019056. A total of 145 adult female mice (Kunming race, albino) aged three to four weeks, were purchased from the animal laboratory of China Medical University. Their weight varied between 22 and 24 g. The temperature and humidity in the animal room ranged from 22 to 24 °C and 50–60%, respectively. Mice were housed five per cage in sterilized plastic cages with wood shaving bedding on a normal 12 h light/dark cycle. Except for exposure time, mice had free access to food and water.

### 2.2. Reagents

TRIzol reagent was purchased from TaKaRa (Nojihigashi, Japan). A BCA protein assay kit, ECL plus kit, and antibodies against ZO-1 were purchased from Thermo Fisher Scientific (Waltham, MA, USA). The primary antibodies against MMP-9 and glial fibrillary acidic protein (GFAP) were obtained from Millipore (Burlington, MA, USA). Primary antibodies against p38, phospho-p38 (p-p38), phospho-NF-κB p65, phospho-inhibitory κBα (p-IκBα), VCAM-1 and β-actin were purchased from Cell Signaling Technology, Inc. (Danvers, MA, USA). Antibodies against IL-1 beta, occludin, claudin 5, ICAM-1, ionized calcium-binding adapter molecule 1 (Iba-1) and iNOS were obtained from Abcam (Cambridge, UK). A goat anti-mouse Alexa Fluor 488 (green), goat anti-rabbit Alexa Fluor 488 (green) and goat anti-mouse Alexa Fluor 594 (red) conjugated secondary as well as a DAB Kit, HRP DAB Detection System, goat serum and DAPI were all products from ZSGB-BIO (Beijing, China). SB202190, SB-3CT and Recombinant Human IL-1ra were obtained from Selleck (Houston, TX, USA) and PeproTech (Allendale, NJ, USA), respectively. RIPA lysis buffer, nuclear and cytoplasmic protein extraction Kit, pyrrolidine dithiocarbamate (PDTC) and biotin-labeled double-stranded oligonucleotide for analysis of NF-κB activity were purchased from Beyotime Biotechnology (Shanghai, China). Duoset IL-1β ELISA kits were obtained from R&D Systems (Minneapolis, MN, USA).

### 2.3. Experimental Procedures and Treatment

Experiments were divided into four parts, which were designed to investigate the specific role of NF-κB, MMP-9, p38 MAPK and IL-1β in 1,2-DCE-induced brain injury. Mice in experimental part one to three were randomly assigned into four groups including a control, 1,2-DCE-intoxicated (intoxicated group) and two intervention groups (pretreated with the low or high dose of respective inhibitor). However, there were five groups in the experimental part four including a control, inhibitor control, intoxicated, and two intervention groups.

Static inhalation exposure was used, and five mice were placed in each static exposure chamber with a capacity of 100 L. A solution of 1,2-DCE with a purity greater than 99% was weighed and added to a plate suspended in the chamber. The initial 1,2-DCE exposure concentration was calculated by the weight of 1,2-DCE divided by the volume of the chamber. After the chamber was sealed, 1,2-DCE was quickly evaporated using a fan in the chamber. The merits of static inhalation exposure are as follows: It is easy to build and operate, and requires consumption of less test chemicals, which is particularly suitable for small animals in the analyses of acute and subacute inhalation exposure. During inhalation exposure, concentrations of oxygen, carbon dioxide and 1,2-DCE in the chamber were examined every hour. At the end of exposure, the humidity and concentrations of oxygen and carbon dioxide in the chamber were lower than 70%, close to 20% and lower than 2%, respectively. The time-weighted average concentrations of 1,2-DCE in the chamber during exposure varied from 1.00 to 1.05 g/m^3^.

After one week of adaption, mice in the intoxicated and intervention groups were exposed to 1.2 g/m**^3^** 1,2-DCE, 3.5 h per day up to three days. Mice in the control and inhibitor control groups were placed in the inhalation chamber without 1,2-DCE for 3.5 h every day. To determine the role of NF-κB in 1,2-DCE-induced brain edema, mice in the intervention groups of experimental part one were intraperitoneally i.p. administered 10 or 100 mg/kg b.w of PDTC in 200 µL saline solution, one hour before every 1,2-DCE exposure. Moreover, to determine the role of MMP-9 in 1,2-DCE-induced BBB disruption, mice in the intervention groups of experimental part two were i.p. administered 12.5 or 25 mg/kg b.w of SB-3CT in 200 µL saline solution with 5% dimethyl sulfoxide (DMSO) and 10% Tween-80, one hour before every 1,2-DCE exposure. In addition, to determine the role of p38 MAPK signaling pathway in 1,2-DCE-induced neuroinflammatory response, mice in the intervention groups of experimental part three were i.p. administered 3.75 and 15 mg/kg b.w of SB202190 in 200 µL saline solution with 50% DMSO, one hour before every 1,2-DCE exposure. Finally, to determine the role of IL-1β in 1,2-DCE-induced activation of p38 MAPK signaling pathway in the brain, mice in the intervention groups of experimental part four were administered by intracerebroventricular (i.c.v) injection 2 or 4 µg of IL-1β receptor antagonist (IL-1ra) in 4 µL sterile saline solution, two hours before every 1,2-DCE exposure. As mentioned above, mice in the control group and intoxicated groups were pretreated with the respective solvent, and mice in the inhibitor control group were i.c.v pretreated with 4 µg IL-1ra.

For i.c.v injection, mice were anesthetized before a small hole in the right parietal bone positioned in a stereotaxic apparatus was drilled. A stainless-steel guide cannula was implanted into the right lateral ventricle (2.5 mm below the skull, 0.8 mm posterior and 1.2 mm lateral to bregma). After securing the implanted cannula with stainless steel screws, it was fixed with dental cement. Operated mice recovered for one week before experiments. The inhibitor solution was injected at a rate of 1.0 µL/min with a syringe (RWD Life Science, China). After injection, the needle was kept for 10 min to allow diffusion of the inhibitor.

All mice in experimental part one to four were anesthetized using 1% pentobarbital sodium 100 mg/kg b.w) and sacrificed by decapitation at 24 h following the three-day exposure. Their brains were taken immediately in a cold plate and kept in −80 °C freezer. Five mice in each group were sacrificed for brain water content, electrophoretic mobility shift assays (EMSA), enzyme-linked immunosorbent assay (ELISA), as well as Western blotting and Real-Time RT-PCR analysis. The others in each group were kept for histological observation and immunohistochemical analysis.

### 2.4. Brain Water Content

Briefly, brains of mice were quickly removed and placed on a dry surface. The cerebral hemisphere was equally cut along sagittal sutures. The left hemispheres were weighed using a chemical balance and then dried in an oven at 100 °C for 48 h. Brain water content was calculated as a percentage using the following formula: [(wet weight − dry weight) / wet weight] × 100%.

### 2.5. Histological Observation

Mice were anesthetized and then perfused transcardially with PBS (pH = 7.4) containing 0.02% heparin for 15 min, followed by 4% paraformaldehyde in PBS for 20 min. Brains were quickly removed and fixed with 4% paraformaldehyde and embedded in paraffin. Serial coronal sections of 5 µm were sliced and collected, and then stained with hematoxylin and eosin (HE staining)

### 2.6. Immunostaining

Following perfusion via the heart, brains were quickly removed. The cerebral samples for immunostaining were immersed in 20% and 30% sucrose for 24 h, and then embedded in OCT-compound. Serial frozen coronal sections (8 μm) were sliced and collected using a cryostat microtome. These were permeabilized for 30 min in PBS containing 0.3% Triton X-100.

In immunofluorescence staining, sections were incubated with normal goat serum, and then with rabbit antibody against VCAM-1 (1:100), and mouse antibodies against GFAP (1:100) at 4 °C overnight in a humidified chamber. After washing with PBS, the goat anti-rabbit and goat anti-mouse secondary antibodies conjugated with Alexa Fluor 488 (green) and Alexa Fluor 594 (red) were added and incubated with the sections for 30 min at 37 °C. Cell nuclei were stained with DAPI for 5 min at room temperature.

In immunohistochemical staining, the sections were treated with hydrogen peroxide to inhibit endogenous peroxidases and then incubated for 30 min with normal goat serum. After, primary antibodies against VCAM-1 (1:50) and Iba-1 (1:100) were added and incubated with the sections overnight. Following incubation with biotinylated secondary antibodies (1:100) at 25 °C for 1 h, the sections were stained using a DAB kit and HRP DAB Detection System, and then counterstained with hematoxylin.

Finally, fluorescence was observed under a fluorescence microscope (Ni; Nikon, Tokyo Japan), and images were captured with a digital camera system (DS-Ri2; Nikon, Tokyo, Japan). Primary antibody incubation was omitted for negative controls.

### 2.7. Western Blotting

Cerebral tissues of the right hemisphere were homogenized in RIPA lysis buffer, and then centrifuged for 20 min at 4 °C, 12,000× *g* to collect the supernatant. Protein concentrations in lysates were determined with a BCA protein assay kit. Total protein at equal concentrations were separated by SDS-PAGE, and then transferred onto a PVDF membrane (Millipore). After blocking with 5% skimmed milk at room temperature, target proteins were probed with primary antibodies against p-p38, p-p65, p-IκB, GFAP, Iba-1, MMP-9, occludin, claudin 5, ZO-1, ICAM-1, VCAM-1, iNOS, IL-1β and β-actin (internal control) at 4 °C overnight. The following day, the membrane was incubated with the secondary antibody conjugated with horseradish peroxidase at room temperature and detected using an ECL plus kit. Membranes were imaged using Azure c300 Chemiluminescent Western Blot Imaging System (Azure Biosystems, Dublin, CA, USA), and assessed using image analyzing software (Gel-Pro analyzer v4.0, Meyer Instruments, Houston, TX, USA). Results were expressed as the relative intensity of the target protein normalized to β-actin (as the internal control) in the cerebral tissues.

### 2.8. Quantitative Real-Time (RT)-PCR

Total RNA in the cerebral tissues was extracted using TRIzol Reagent. The first-strand cDNA was synthesized from total RNA using the PrimeScript RT reagent kit (Takara, Japan). To amplify a fragment of ICAM-1, VCAM-1, iNOS, IL-1β and GAPDH (used as an internal control), the following specific primer pairs detailed in Table 1 were used. Amplification was conducted using the SYBR Premix Ex TaqII (Takara, Nojihigashi, Japan) and a QuantStudio 6 Flex real-time PCR System (Life Technologies, Carlsbad, CA, USA) for 40 cycles of 5 s at 95 °C and 34 s at 60 °C. Results were evaluated using the comparative Ct method. RNA abundance was expressed as 2^−ΔΔCt^ for the target gene normalized to the GAPDH gene (as the internal control) and presented as fold change versus contralateral control samples.

### 2.9. Electrophoretic Mobility Shift Assay (EMSA)

Nuclear protein was extracted and quantified following the protocol of the nuclear and cytoplasmic protein extraction kit, and EMSA was performed according to manufacturer’s instructions of Chemiluminescent EMSA Kit (Beyotime Institute of Biotechnology, Shanghai, China). Briefly, biotin-labeled oligonucleotide probe of NF-κB (5’-AGT TGA GGG GAC TTT CCC AGG C-3’) was incubated with the nuclear protein at room temperature for 20 min, and then the mixture was subjected to nondenaturing 4% polyacrylamide gel electrophoresis in a 0.5 × Tris-borate-EDTA buffer. Samples were transferred to a positively charged nitrocellulose membrane and cross-linked under UV light. Finally, levels of NF-κB DNA binding activity were quantified using a Streptavidin-HRP conjugated chemiluminescent substrate.

### 2.10. ELISA

Levels of IL-1β in the lysates of mouse brains were also determined using the specific ELISA kit according to the manufacturer’s instructions. Contents of IL-1β in the brain samples were expressed as pg per milligram protein.

### 2.11. Statistical Analysis

Software SPSS for Windows v13.0 (SPSS Inc. IL, SPSS, Armonk, NY, USA) was used for statistical analysis. All data were expressed as mean ± standard deviations (SD), and analysis of variance test (one-way ANOVA) followed by Student–Newman–Keuls test (SNK) was carried out to evaluate the significant differences among groups. Statistical significance was accepted at *p* less than 0.05.

## 3. Results

### 3.1. Involvement of NF-κB in 1,2-DCE-Induced Brain Edema in Mice

Consistent with our previous studies [7,25], brain edema formed in the brain of mice in the 1,2 DCE-intoxicated group, which was indicated by increased brain water content and morphological changes of brain edema (Figure 1A,B). Moreover, compared with the control group, NF-κB binding activities in the brain of mice increased dramatically in the intoxicated group (Figure 1C), suggesting that NF-κB was activated during the course of 1,2-DCE-induced brain edema. On the other hand, pretreatment of 1,2-DCE-intoxicated mice with PDTC, a specific inhibitor of NF-κB, dose-dependently attenuated the changes in NF-κB binding activities, brain water content, and pathological observation of brain edema, suggesting that activation of NF-κB was involved in 1,2-DCE-induced brain edema in mice. Pretreatment of 1,2-DCE-intoxicated mice with PDTC also dose-dependently attenuated the changes observed in overexpression of MMP-9 (Figure 2A–C) and decreased protein levels of ZO-1, occludin and claudin 5 (Figure 2D,E), suggesting that activation of NF-κB might contribute to MMP-9 overexpression and BBB disruption in the brain of 1,2-DCE-intoxicated mice.

Results of the present study also revealed that the expression levels of VCAM-1, ICAM-1, iNOS, and IL-1β were enhanced transcriptionally during the course of 1,2-DCE-induced brain edema, whereas pretreatment of 1,2-DCE-intoxicated mice with PDTC attenuated changes in expression of these inflammatory mediators and the proinflammatory factor (Figure 3A–D). Additionally, our results showed that compared to the control group, protein levels of GFAP and Iba-1 and number of Iba-1 positive cells in the brain of mice increased significantly in the intoxicated group, where pretreatment with PDTC could dose-dependently attenuate these changes (Figure 4A–D). Taken together, the present study demonstrated for the first time that activation of glial cells, and overexpression of proinflammatory factors were mediated through the NF-κB signaling pathway during the course of 1,2-DCE-induced brain edema in mice.

### 3.2. MMP-9 Involvement in 1,2-DCE-Induced BBB Disruption in Mice

Consistent with our previous studies [6,7], MMP-9 was overexpressed and BBB was disrupted during the course of 1,2-DCE-induced brain edema in mice. Moreover, we further demonstrated that MMP-9 overexpression and BBB disruption during the course of 1,2-DCE-induced brain edema were mediated through the p38 MAPK/NF-κB signaling pathway. To determine the association between MMP-9 overexpression and BBB disruption, SB-3CT, a specific inhibitor of MMP-9, was selected in this part of the experiment. As expected, pretreatment with SB-3CT abolished the decrease in protein levels of ZO-1, occludin and claudin 5 in 1,2-DCE-intoxicated mice (Figure 5A,B). Thus, it is reasonable to conclude that BBB disruption during the course of 1,2-DCE-induced brain edema could result from excessive MMP-9 mediated through activation of the p38 MAPK/ NF-κB signaling pathway.

### 3.3. p38 MAPK Signaling Pathway Involvement in the Inflammatory Reaction During the Course of 1,2-DCE-Induced Brain Edema in Mice

As mentioned previously, activation of microglia and astrocytes and overexpression of the pro-inflammatory factor IL-1β and inflammatory mediators were mediated via NF-κB signaling pathway during the course of 1,2-DCE-induced brain edema in mice. To further confirm the role of the p38 MAPK signaling pathway in 1,2-DCE-induced neuroinflammatory reaction, SB202190, a specific inhibitor against p38 MAPK was used. Pretreatment of 1,2-DCE-intoxicated mice with SB202190 dose-dependently attenuated changes in protein levels of p-p65, Iba-1, GFAP, VCAM-1, ICAM-1, iNOS, and IL-1β (Figure 6). Therefore, these results confirmed that activation of glial cells and overexpression of the pro-inflammatory factor and inflammatory mediators during the course of 1,2-DCE-induced brain edema in mice were mediated through the p38 MAPK/ NF-κB signaling pathway.

### 3.4. IL-1β Involvement in the Neuroinflammatory Reaction During the Course of 1,2-DCE-Induced Brain Edema in Mice

As aforementioned, IL-1β is overexpressed through the activation of p38 MAPK/NF-κB signaling pathway during the course of 1,2-DCE-induced brain edema in mice. However, whether IL-1β also plays a role in activation of p38 MAPK/ NF-κB signaling pathway during the course of 1,2-DCE-induced brain edema in mice is not known. To answer this question, IL-1ra, a specific antagonist of IL-1β receptor was used. Consistent with previously mentioned data, the expression of IL-1β in the brain of mice in the intoxicated group was transcriptionally upregulated along with the exposure days, and increased significantly after two days of exposure, as compared to the control group (Figure 7A,B). Moreover, pretreatment with IL-1ra in 1,2-DCE-intoxicated mice attenuated the expression of IL-1β at the transcriptional level (Figure 7C,D), suggesting that IL-1β promotes its own expression. Furthermore, pretreatment with IL-1ra also attenuated p-p38, p-p65, p-IκB, VCAM-1, ICAM-1 and Iba-1 protein levels (Figure 8), suggesting that IL-1β could promote the expression of the inflammatory mediators and activation of microglia through p38 MAPK/ NF-κB signaling pathway. However, IL-1β might not be directly involved in overexpression of iNOS, and activation of astrocytes during the course of 1,2-DCE-induced brain edema in mice. In addition, all mentioned indicators in the brain of mice did not differ significantly between the blank control and inhibitor control groups.

## 4. Discussion

Several novel findings were uncovered in this study. First, we confirmed that overexpression of MMP-9 in the brain of 1,2-DCE-intoxicated mice was mediated by p38 MAPK/ NF-κB signaling pathway and contributed to the disruption of the BBB during the course of brain edema. Second, neuroinflammatory reactions indicated by activation of glial cells, and enhanced production of inflammatory mediators, was triggered by activation of the p38 MAPK/ NF-κB signaling pathway during the course of 1,2-DCE-induced brain edema in mice. In addition, IL-1β also promoted the expression of itself and the inflammatory mediators, as well as activation of microglia through the p38 MAPK/ NF-κB signaling pathway during the course of 1,2-DCE-induced brain edema in mice. These findings suggested that the positive feedback loop between activation of p38 MAPK/ NF-κB signaling pathway and production of the proinflammatory factor IL-1β may amplify neuro-inflammatory reactions and exacerbate the brain injury during the course of 1,2-DCE-induced brain edema.

A neuroinflammatory reaction is a complex biological process in response to intoxication, infection, and trauma, involving all the cells in the brain, such as neurons, astrocytes, and microglia [26], which may activate microglia and astrocytes, promote the production and release of proinflammatory factors and inflammatory mediators, disrupt BBB integrity and recruit peripheral immune cells [27]. Additionally, acute neuroinflammatory reactions can be induced by rapid and early activation of the glial cells as a response to cellular irritants. Microglia, as the main innate immune cells within the brain, are very reactive cells. Any changes in the brain immediately lead to their activation, proliferation, and morphological changes [28]. Thus, microglia are the first line of defense against brain injury [29,30]. Iba-1 is specifically expressed in microglia and widely used as a marker for microglia activation [31]. When activated, microglia can release and produce large amounts of pro-inflammatory factors to induce neurotoxicity [32,33,34]. Astrocytes are multifunctional cells that control the ion homeostasis, oxidative stress and blood flow in the brain, as well as participate in the formation and functioning of BBB [35]. Like microglia, astrocytes are activated by pathological factors that change their morphology and functions. Activation of astrocytes is characterized by three hallmarks, including GFAP elevation, hypertrophy, and increased proliferation [36]. Most importantly, activation of astrocytes and microglia leads to the release of proinflammatory factors, which then activate more astrocytes and microglia, and result in the production and secretion of more proinflammatory factors. This feedback loop creates and propels neuroinflammatory reactions that lead to brain injury. In the present study, we found that the number of activated microglia, and protein levels of Iba-1 and GFAP increased in the brain of 1,2-DCE-intoxicated mice, and pretreatment with either PDTC or SB202190 inhibitors attenuates these changes. Thus, our results indicated that subacute poisoning of 1,2-DCE could activate microglia and astrocytes through the p38 MAPK/NF-κB signaling pathway.

NF-κB, as a transcription factor can be activated by several signaling pathways, including phosphoinositide 3-kinase/protein kinase B (PI3K/AKT) and mitogen-activated protein kinases (MAPKs), and mediates inflammatory reactions through the production of more pro-inflammatory cytokines, chemokines and inducible enzymes [13,37,38]. The common form of NF-κB consists of p50 (NF-κB1) and p65 (RelA) and remains in the cytoplasm as an inactive form through interaction with the NF-κB inhibitory protein IκB. When phosphorylated, IκB is degraded by the proteasome and then liberated NF-κB is translocated from the cytoplasm to the nucleus where it initiates the transcription of target genes [39]. Findings from our previous work have revealed that the p38 MAPK signaling pathway was activated and involved in NF-κB activation and MMP-9 overexpression during the course of 1,2-DCE-induced brain edema in mice [6]. In the present study, we found that pretreatment of 1,2-DCE-intoxicated mice with PDTC also transcriptionally attenuates MMP-9 overexpression [40], meanwhile ameliorating disruption of TJs and changes of brain edema in these mice. Thus, we confirmed that p38 MAPK/ NF-κB signaling pathway plays an important role in MMP-9 overexpression, BBB disruption, and brain edema formation in 1,2-DCE-intoxicated mice.

MMP-9 belongs to a family of extracellular proteases, normally expressed at low levels but substantially overexpressed in many brain diseases [41]. In the brain, MMP-9 not only acts as an inflammatory mediator involved in the development of inflammation reactions, but also as a proteolytic enzyme involved in BBB degradation [8,42]. Studies to date have demonstrated that excessive MMP-9 in the brain may aggravate inflammatory reactions and allows for inflammation infiltration into the brain parenchyma since it disrupts the integrity of BBB by attacking the extracellular matrix, basal lamina and TJs in the BBB [10,43,44]. Moreover, MMP-9 is also involved in the cleavage of IL-1β from its precursor to mature form [9]. On the other hand, TJs, commonly composed of ZO-1, occludin, and claudin5, are known to be important structural components, and necessary for the integrity of BBB [45]. The present study together with our previous work revealed that the levels of ZO-1, occludin, and claudin5 in the brain of 1,2-DCE-intoxicated mice decreased markedly during the course of brain edema [6], and pretreatment with SB-3CT could abolish the changes in expression of TJs. Thus, our studies confirmed that disrupted expression of TJs in the brain of 1,2-DCE-intoxicated mice was due to excessive MMP-9. Taken together, these indicated that MMP-9 overexpression was mediated by the p38 MAPK/ NF-κB signaling pathway, and in turn resulted in disruption of BBB integrity, causing the formation of brain edema during the course of 1,2-DCE-induced brain edema in mice.

In the brain, IL-1β is produced primarily by activated microglia, and then released into the extracellular space, where it is activated by MMP-9, and binds to its specific receptor in glial and endothelial cells to stimulate neuroinflammatory reactions [46,47]. It has been reported that intracerebral injection of IL-1β in rats might facilitate BBB breakdown and inflammation infiltration into the brain [48]. In addition, IL-1β could induce MMP-9 gene expression in astrocytes and enhance the expression of VCAM-1 and ICAM-1 in mouse brain endothelium via the MAPKs/ NF-κB signaling pathways [49,50]. In the current study, IL-1β in the brain of 1,2-DCE-intoxicated mice was transcriptionally upregulated, and the changes were attenuated by pretreatment with IL-1ra, SB202190 or PDTC. Since IL-1ra competitively binds to the receptor of IL-1β, it blocks the biological activity of IL-1β rather than directly reducing its protein levels. Thus, our results suggested that induction of IL-1β in the brain of 1,2-DCE-intoxicated mice was partly due to IL-1β itself, and there was a positive feedback loop between production of IL-1β and activation of p38 MAPK/ NF-κB signaling pathway. On the other hand, pretreatment of 1,2-DCE-intoxicated mice with IL-1ra, or SB202190 or PDTC could also attenuate the increase in expression of VCAM-1 and ICAM-1. As cell-adhesion molecules, both VCAM-1 and ICAM-1 are expressed on the luminal surface of endothelial cells in BBB and appear to be important mediators in the neuroinflammatory reactions. Under steady-state conditions, the expression levels of VCAM-1 and ICAM-1 are usually low. It has been reported that down-regulated expression of VCAM-1 and ICAM-1 could protect against disruption of BBB [20,21,51]. Findings from the current study showed that ICAM-1 and VCAM-1 were transcriptionally upregulated in the brain of 1,2-DCE-intoxicated mice through the p38 MAPK/ NF-κB signaling pathway, and also being induced by IL-1β.

It is well established that iNOS in the brain is not normally expressed but appears in response to inflammatory stimuli in several pathological states including cerebral edema [52,53]. The soluble gas nitric oxide (NO) is crucial for a variety of normal physiological functions concerned with the maintenance of the CNS. When produced in excessive amounts due to induction of iNOS, the large quantities of NO become neurotoxic, and are usually linked to a variety of pathologies and subsequent opening of the BBB, leading to vasogenic brain edema and subsequent secondary brain damage [54]. Moreover, NO can react with superoxide anion to form peroxynitrite that as a powerful oxidant can cause extensive cellular damage by oxidizing proteins, lipids, and DNA. Furthermore, the mitochondrial electron transport chain can be impaired by excessive NO, and therefore diminishes cellular energy production, leading to ATP depletion and neuronal cell death. In the present study, iNOS expression increased dramatically in the brain of 1,2-DCE-intoxicated mice, and its changes were attenuated by pretreatment with either SB202190 or PDTC. However, the changes in protein levels of iNOS and GFAP were insensitive to the IL-1β receptor antagonist.

## 5. Conclusions

In conclusion, our findings demonstrate that subacute poisoning of 1,2-DCE could stimulate microglia and astrocytes to release and produce more of the proinflammatory factor IL-1β and inflammatory mediators, including MMP-9, VCAM-1, ICAM-1, and iNOS, through the p38 MAPK/ NF-κB signaling pathway, which further induce neuroinflammatory reactions that contribute to the disruption of TJs, increase of BBB permeability and infiltration of peripheral immune cells into the brain, finally leading to brain edema in 1,2-DCE-intoxicated mice. The proposal schematic diagram was shown in Figure 9. Hence, inhibiting neuroinflammatory reactions would be an effective therapeutic approach in order to mitigate the progression of brain edema induced by subacute poisoning of 1,2-DCE [55].

## Figures and Tables

**Figure 1 cells-08-00987-f001:**
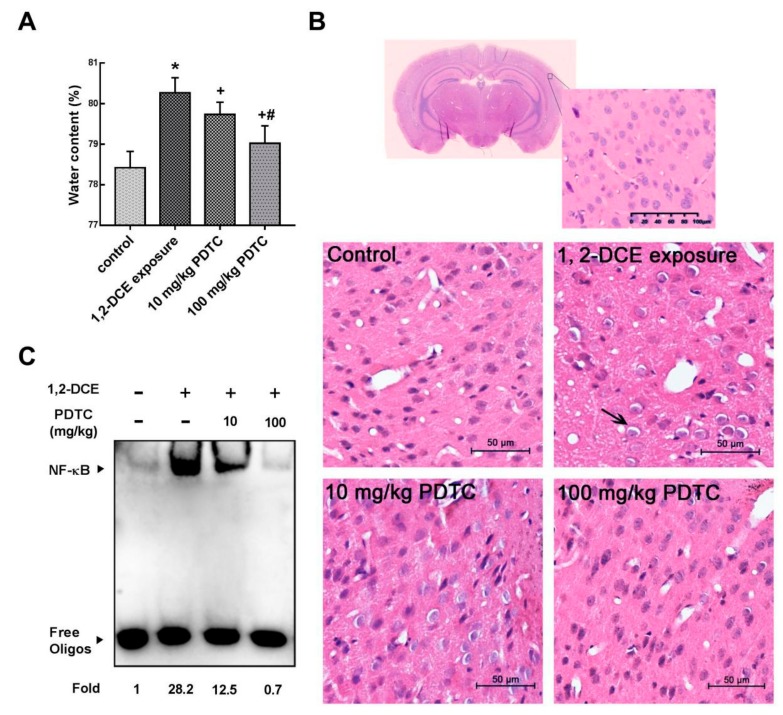
Involvement of NF-κB in brain edema in 1,2-DCE-intoxicated mice. (**A**) Comparison of mouse brain water content among groups. (**B**) The photomicrographs of HE staining in the frontoparietal region of the cerebral cortex are representative of five separate experiments and were captured using a Nikon microscope (200×). The arrow indicates the enlarged perinuclear space. Scale bar represents 50 μm. (**C**) The image of DNA binding activity of NF-κB by electrophoretic mobility shift assay (EMSA) is representative of at least three experiments. Data expressed as mean ± SD are the results of five independent experiments and analyzed by one-way ANOVA. Significant difference is defined as *p* < 0.05, and *, vs. control group; ^+^, vs. 1,2-DCE poisoned group; ^#^ vs. low dose intervention group.

**Figure 2 cells-08-00987-f002:**
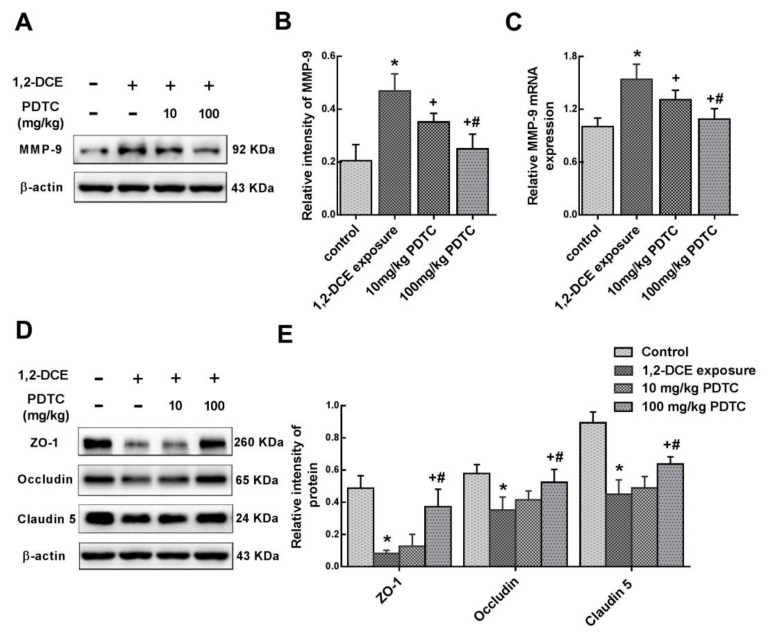
Involvement of NF-κB in MMP-9 overexpression and TJs disruption in 1,2-DCE-intoxicated mice. (**A**,**D**) Representative images of Western blot analysis. (**B**,**E**) Graphs show the densitometric quantitation of MMP-9, ZO-1, occludin and claudin-5, which were normalized to the intensity of β-actin. (**C**) Measurement of mRNA levels by real-time RT-PCR. The mRNA levels of MMP-9 were normalized to GAPDH and presented as fold change of the control group. Data expressed as mean ± SD are the results of five independent experiments and analyzed by one-way ANOVA. Significant difference is defined as *p* < 0.05, and *, vs. control group; ^+^, vs. 1,2-DCE poisoned group; ^#^ vs. low dose intervention group.

**Figure 3 cells-08-00987-f003:**
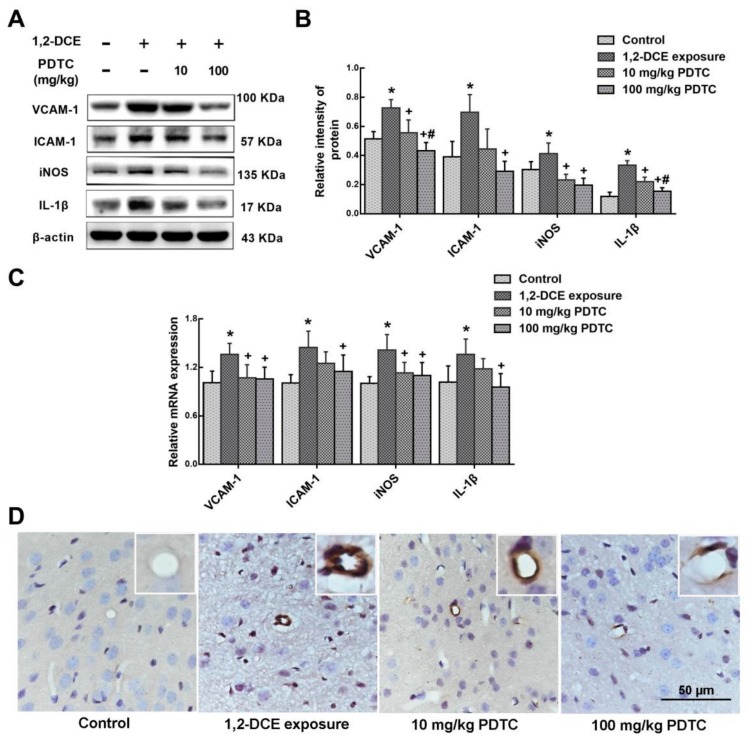
Involvement of NF-κB in upregulation of VCAM-1, ICAM-1, iNOS, and IL-1β expression in 1,2-DCE-intoxicated mice. (**A**) Representative images of Western blot analysis. (**B**) Graphs show the densitometric quantitation of VCAM-1, ICAM-1, iNOS, and IL-1β, which were normalized to the intensity of β-actin. (**C**) Measurement of mRNA levels by real-time RT-PCR. The mRNA levels of VCAM-1, ICAM-1, iNOS, and IL-1β were normalized to GAPDH and presented as fold change of the control group. (**D**) The photomicrographs of immunocytochemistry represent staining of VCAM-1 in the frontoparietal region of the cerebral cortex, which are representative of five separate experiments and were captured using a Nikon microscope (200×). Scale bar represents 50 μm. Data expressed as mean ± SD are the results of five independent experiments and analyzed by one-way ANOVA. Significant difference is defined as *p* < 0.05, and *, vs. control group; ^+^, vs. 1,2-DCE poisoned group; ^#^ vs. low dose intervention group.

**Figure 4 cells-08-00987-f004:**
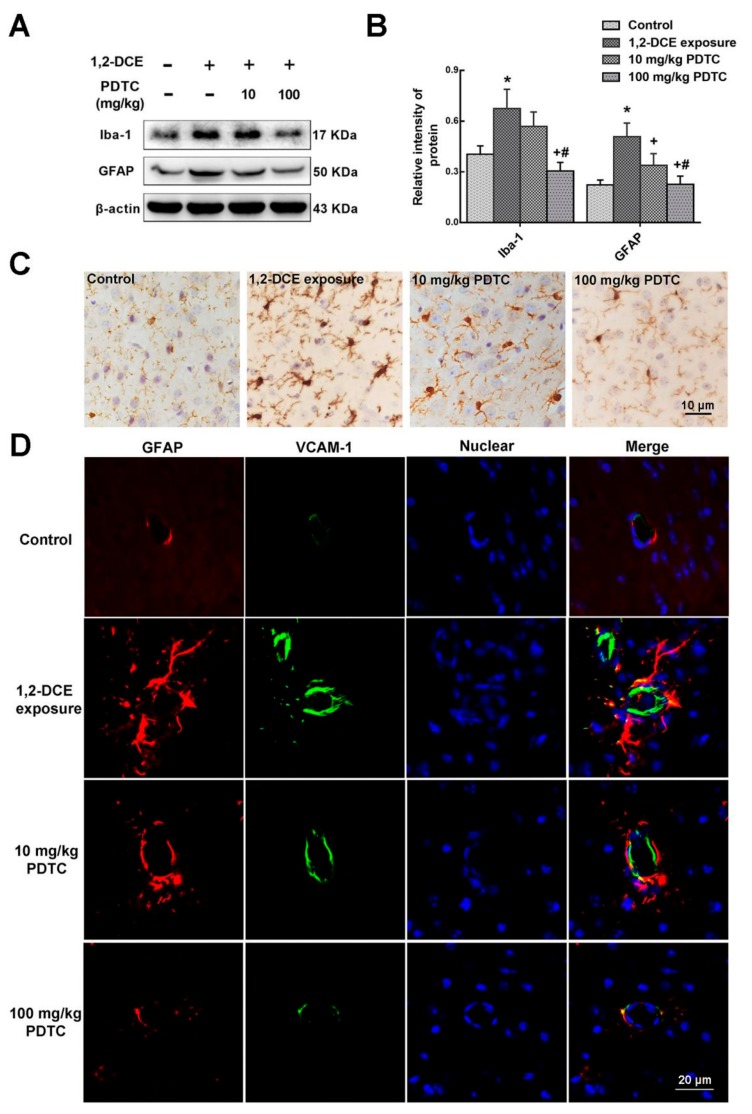
Involvement of NF-κB in upregulation of Iba-1 and GFAP expression in 1,2-DCE-intoxicated mice. (**A**) Representative images of Western blot analysis. (**B**) Graphs show the densitometric quantitation of Iba-1 and GFAP, which were normalized to the intensity of β-actin. (**C**,**D**) The photomicrographs of immunocytochemistry and immunofluorescence represent staining of Iba-1, GFAP, and VCAM-1 in the frontoparietal region of the cerebral cortex, respectively, which are representative of five separate experiments and were captured using a Nikon microscope (200×). Scale bar represents 10 μm and 20 μm, respectively. Data expressed as mean ± SD are the results of five independent experiments and analyzed by one-way ANOVA. Significant difference is defined as *p* < 0.05, and *, vs. control group; ^+^, vs. 1,2-DCE poisoned group; ^#^ vs. low dose intervention group.

**Figure 5 cells-08-00987-f005:**
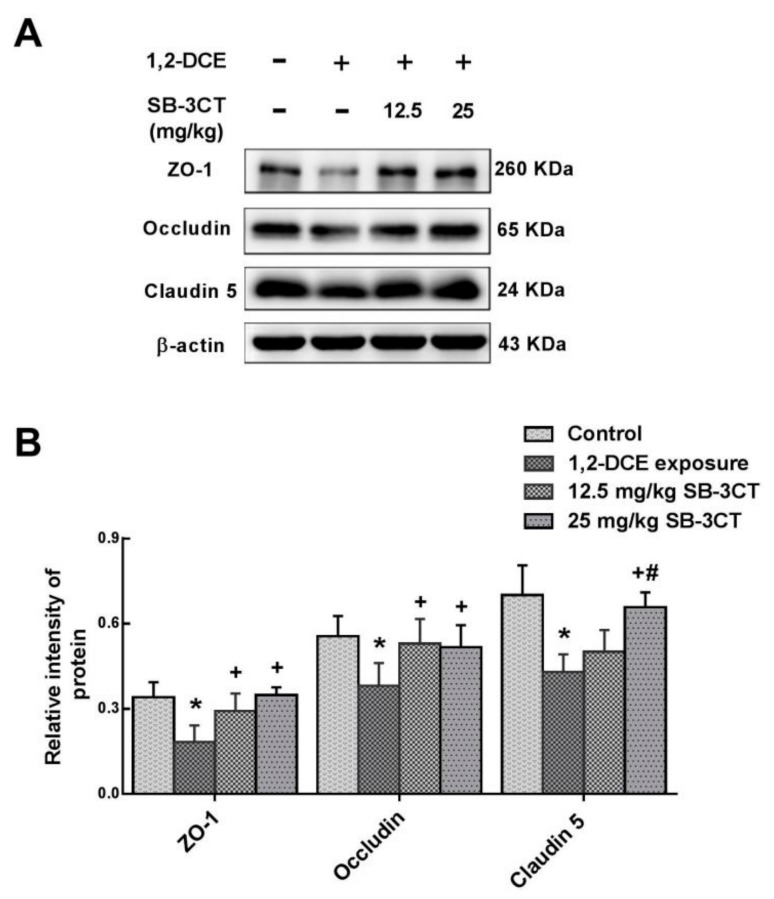
Involvement of MMP-9 in the decrease of TJs expression in 1,2-DCE-intoxicated mice. (**A**) Representative images of Western blot analysis. (**B**) Graphs show the densitometric quantitation of ZO-1, occludin, and claudin-5, which were normalized to the intensity of β-actin. Data expressed as mean ± SD are the results of five independent experiments and analyzed by one-way ANOVA. Significant difference is defined as *p* < 0.05, and *, vs. control group; ^+^, vs. 1,2-DCE poisoned group; ^#^ vs. low dose intervention group.

**Figure 6 cells-08-00987-f006:**
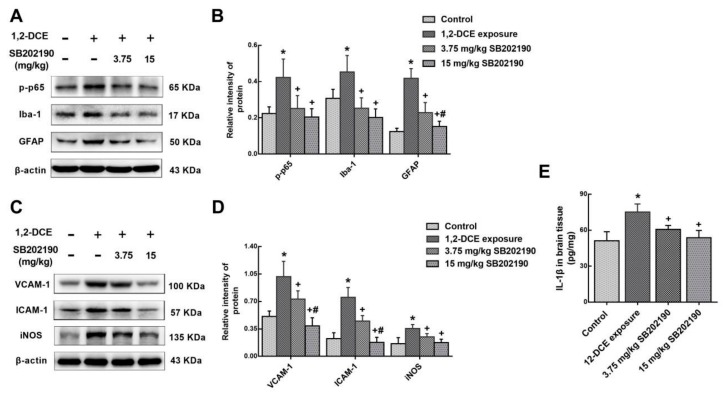
Involvement of p38 in upregulation of p-p65, Iba-1, GFAP, VCAM-1, ICAM-1, iNOS, and IL-1β expression in 1,2-DCE-intoxicated mice. (**A**,**C**) Representative images of Western blot analysis. (**B**,**D**) Graphs show the densitometric quantitation of p-p65, Iba-1, GFAP, VCAM-1, ICAM-1, and iNOS, which were normalized to the intensity of β-actin. (**E**) IL-1β expression levels in the brain tissues measured by ELISA. Data expressed as mean ± SD are the results of five independent experiments and analyzed by One-way ANOVA. Significant difference is defined as *p* < 0.05, and *, vs. control group; ^+^, vs. 1,2-DCE poisoned group; ^#^ vs. low dose intervention group.

**Figure 7 cells-08-00987-f007:**
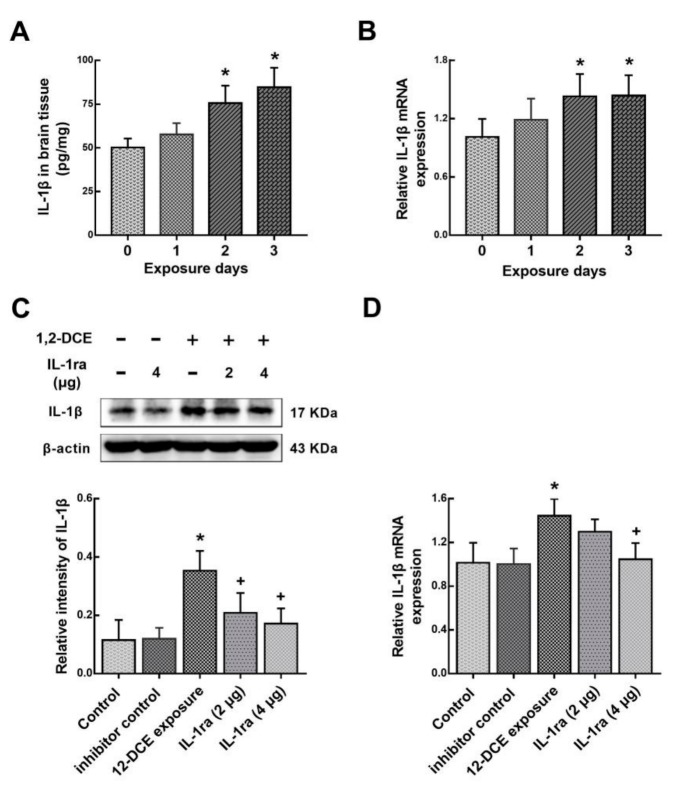
Changes of IL-1β along with the exposure days and involvement of IL-1ra in IL-1β expression in 1,2-DCE-intoxicated mice. (**A**) IL-1β expression levels in the brain tissues measured by ELISA. (**B**,**D**) Measurement of mRNA levels by real-time RT-PCR. The mRNA levels of IL-1β were normalized to GAPDH and presented as fold change of the control group. (**C**) Images represent Western blot, and graphs show the densitometric quantitation of IL-1β, which were normalized to the intensity of β-actin. Data expressed as mean ± SD are the results of five independent experiments and analyzed by one-way ANOVA. Significant difference is defined as *p* < 0.05, and *, vs. control group; ^+^, vs. 1,2-DCE poisoned group; ^#^ vs. low dose intervention group.

**Figure 8 cells-08-00987-f008:**
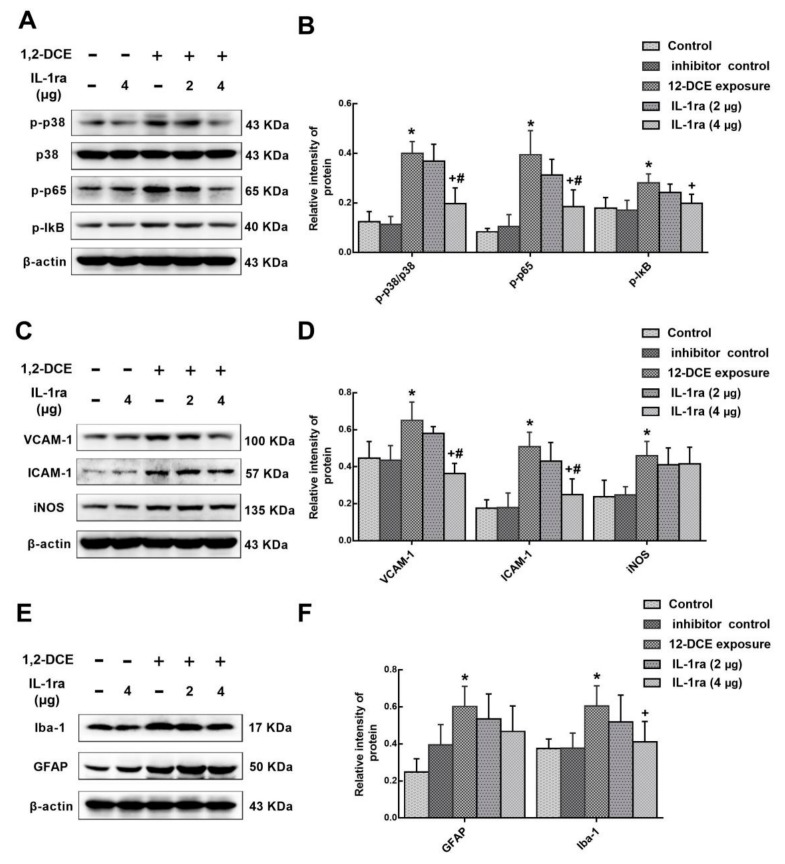
Involvement of IL-1ra in expression of p-p38, p-p65, p-IκB, VCAM-1, ICAM-1, iNOS, Iba-1 and GFAP in 1,2-DCE-intoxicated mice. (**A**,**C**,**E**) Representative images of Western blot analysis. (**B**,**D**,**F**) Graphs show the densitometric quantitation of p-p38, p-p65, p-IκB, VCAM-1, ICAM-1, iNOS, Iba-1 and GFAP, which were normalized to the intensity of β-actin. Data expressed as mean ± SD are the results of five independent experiments and analyzed by one-way ANOVA. Significant difference is defined as *p* < 0.05, and *, vs. control group; ^+^, vs. 1,2-DCE poisoned group; ^#^ vs. low dose intervention group.

**Figure 9 cells-08-00987-f009:**
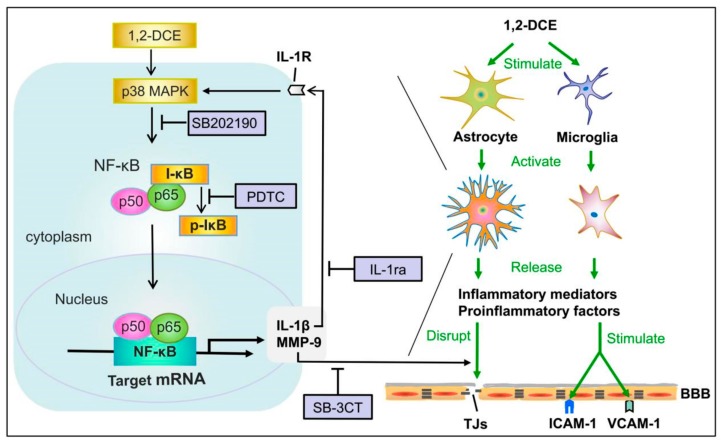
Schematic diagram.

**Table 1 cells-08-00987-t001:** The sequence of primer pairs for RT-PCR analysis.

Gene	Primer (5′→3′)	Primer Sequences	Length (bp)
*Mmp9* (MMP-9)	Sense	GAAGGCTCTGCTGTTCAG	129
	Antisense	AAGATGTCGTGTGAGTTCC	
*Vcam1* (VCAM-1)	Sense	CTGTTCCAGCGAGGGTCTAC	287
	Antisense	CACAGCCAATAGCAGCACAC	
*Icam1* (ICAM-1)	Sense	GTGGGTCGAAGGTGGTTCTT	168
	Antisense	GCAGTTCCAGGGTCTGGTTT	
*Nos2* (iNOS)	Sense	GGGTCACAACTTTACAGGGAGT	149
	Antisense	GAGTGAACAAGACCCAAGCG	
*Il1**β* (IL-1β)	Sense	GAAATGCCACCTTTTGACAGTG	116
	Antisense	TGGATGCTCTCATCAGGACAG	
*GAPDH*	Sense	CAATGTGTCCGTCGTGGATCT	124
	Antisense	GTCCTCAGTGTAGCCCAAGATG	

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
