# Peer review of "Neuroinflammatory Reactions in the Brain of 1,2-DCE-Intoxicated Mice during Brain Edema"

_cells, 2019, doi:10.3390/cells8090987_

Round 1

Reviewer 1 Report

This study deals about the mechanism of brain edema induced by1,2-DCE exposure in mice. On the basis of their experiments the authors state that 1,2-DCE exposure stimulate astrocyte and microglia, which induce production of inflammatory mediators and promote BBB disruption. Although the study may be potentially interesting, in my opinion there are some issues that should be clarified.

1) Does inhibitor alone alter brain water content and protein expression levels of ZO-1, Occludin and Claudin 5? The authors should show the results of inhibitor-only control.

2) Aquaporin (AQP) 4 expression in astrocyte is deeply involved in the formation and progression of brain edema. The authors need to test AQP4 expression levels in the brain with 1,2-DCE exposure (if possible).

3) The photomicrograph in Figure 1B, Figure 3D and Figure 4C, D (a-d) are difficult to understand. I think it’s better to show the group name (Control, 10 mg/kg PDTC….) in Figure, not a, b, c and d.

4) Figure 1B show cerebral cortex. Which brain part is shown in Figure 3D, Figure 4C, D? Especially, the density of microglia differs at each brain regions. The authors should show the brain regions in Figure 3D, Figure 4C, D.

5) The authors suggested that 1,2-DCE promote BBB disruption and PDTC inhibit this disruption. The authors should investigate the permeability of BBB using Evans Blue.

6) GFAP is a typical marker of astrocyte. But, in figure 4D-a is slightly stained with GFAP. Please explain this contradiction.

7) The abbreviation of figure 5B (OC and C5) should be explained in figure legend.

8) The authors need to further explain about the knowledge of the relationship between brain edema and iNOS in Introduction or Discussion section.

Author Response

This study deals about the mechanism of brain edema induced by 1,2-DCE exposure in mice. On the basis of their experiments the authors state that 1,2-DCE exposure stimulate astrocyte and microglia, which induce production of inflammatory mediators and promote BBB disruption. Although the study may be potentially interesting, in my opinion there are some issues that should be clarified.

1) Does inhibitor alone alter brain water content and protein expression levels of ZO-1, Occludin and Claudin 5? The authors should show the results of inhibitor-only control.

In the previous study (PMID: 30081051), we have reported that treatment with the inhibitor of p38 MAPK alone did not alter the brain water content, and protein expression and phosphorylation of p38 MAPK in the control mice.

2) Aquaporin (AQP) 4 expression in astrocyte is deeply involved in the formation and progression of brain edema. The authors need to test AQP4 expression levels in the brain with 1,2-DCE exposure (if possible).

Indeed, in the previous study (PMID: 24964198), we found that exposure to 1,2-DCE might up-regulate the expression of AQP4 protein at the early phase of brain edema, suggested that AQP4 might be involved in the formation and progression of 1,2-DCE induced brain edema.

3) The photomicrograph in Figure 1B, Figure 3D and Figure 4C, D (a-d) are difficult to understand. I think it’s better to show the group name (Control, 10 mg/kg PDTC….) in Figure, not a, b, c and d.

According to this suggestion, the group names in the Figures have been changed in the revised manuscript.

4) Figure 1B show cerebral cortex. Which brain part is shown in Figure 3D, Figure 4C, D? Especially, the density of microglia differs at each brain regions. The authors should show the brain regions in Figure 3D, Figure 4C, D.

Figure 3D and Fig. 4C, D showed the frontoparietal region in cerebral cortex. According to this suggestion, the brain region shown in Fig. 3D and Fig. 4C, D have been indicated in the revised manuscript.

5) The authors suggested that 1,2-DCE promote BBB disruption and PDTC inhibit this disruption. The authors should investigate the permeability of BBB using Evans Blue.

In our previous study (PMID: 30081051), the BBB permeability has been checked by using sodium fluorescein, since the molecular weight of sodium fluorescein (376.27) is much smaller than Evans Blue (960.81). Our previous results showed that the BBB permeability increased along with the exposure days in 1,2-DCE-intoxicated mice. By the way, we failed to observe any changes in BBB permeability checked by Evans Blue as the tracer in 1,2-DCE-intoxicated mice.

6) GFAP is a typical marker of astrocyte. But, in figure 4D-a is slightly stained with GFAP. Please explain this contradiction.

As shown in Fig. 4A, the GFAP protein is expressed detectably in the brain of control mice, and the expression levels of GFAP protein were enhanced due to activation of astrocytes in 1,2-DCE intoxicated mice. However, when we examined GFAP expression by immunofluorence staining, if the image of control group in Fig. 4D revealed clearly, the images of treatment groups would be too strong to be seen clearly in Fig 4D due to capturing under the same filming conditions.

7) The abbreviation of figure 5B (OC and C5) should be explained in figure legend.

According to this suggestion, “OC and C5” labeled under the x-axis of Fig. 5B has been changed, and the full name of the proteins are given in the revised manuscript.

8) The authors need to further explain about the knowledge of the relationship between brain edema and iNOS in Introduction or Discussion section.

According to this suggestion, further explain about the relationship between brain edema and iNOS have been added in the section of Discussion in the revised manuscript.

Reviewer 2 Report

Cells_576909_comments

Neuroinflammatory Reactions in the Brain of 1,2-DCE-IntoxicatedMice During Brain Edema by Xiaoxia Jin et al.

The paper is well written and the tests are carefully planned and performed. The authors show and conclude that in 1,2-dichloroethane induced brain edema in mice the p38 MAPK/ NF-κB signaling pathway might be involved in the activation of glial cells, and that the overproduction of proinflammatory factors might induce inflammatory reactions. However, their reference to a practical application in which inhibiting neuroinflammatory reactions would be effective therapeutic approach for brain edema is not a novel finding. Thus the significance of the results of the work for practical nursing remains open. This part could be clarified more in the conclusions section.

Comments:

Please explain the abbreviation “p38MAPK” when first mentioned. Please check also the other abbreviations. Please correct the superscript for cubic meter. (Page 3) Please use μL instead of μl. (Pages 3 and 4) Please explain the total number of mice used in the study. (Page 2) Please explain in detail what kind of pain medication was used after i.c.v injection. Figure 1. “Arrows indicate…”. Only one arrow can be found in the panel B.

Author Response

Neuroinflammatory Reactions in the Brain of 1,2-DCE-Intoxicated Mice During Brain Edema by Xiaoxia Jin et al.

The paper is well written and the tests are carefully planned and performed. The authors show and conclude that in 1,2-dichloroethane induced brain edema in mice the p38 MAPK/ NF-κB signaling pathway might be involved in the activation of glial cells, and that the overproduction of proinflammatory factors might induce inflammatory reactions. However, their reference to a practical application in which inhibiting neuroinflammatory reactions would be effective therapeutic approach for brain edema is not a novel finding. Thus the significance of the results of the work for practical nursing remains open. This part could be clarified more in the conclusions section.

Comments:

1) Please explain the abbreviation “p38 MAPK” when first mentioned. Please check also the other abbreviations.

According to this suggestion, the full names of abbreviations have been given, when they are first mentioned in the revised manuscript.

2) Please correct the superscript for cubic meter. (Page 3)

According to this suggestion, the superscript for cubic meter has been corrected in the revised manuscript.

3) Please use μL instead of μl. (Pages 3 and 4)

According to this suggestion, All “μl” have been replaced by “μL” in the revised version.

4) Please explain the total number of mice used in the study. (Page 2)

In the present study, total 145 mice were used. According to this suggestion, the total number of mice used in this study has been given in the revised manuscript.

5) Please explain in detail what kind of pain medication was used after i.c.v injection.

 For i.c.v injection, mice were anesthetized, and then a small hole in the right parietal bone positioned in a stereotaxic apparatus was drilled. A stainless-steel guide cannula was implanted into the right lateral ventricle on a stereotaxic apparatus. After securing the implanted cannula with stainless steel screws, it was fixed with dental cement. Operated mice recovered for one week before experiments. The mice were also anesthetized with pentobarbital sodium, when the inhibitor solution was injected. After injection, the mice were under anesthesis for more than one hour according to the experimental protocols introduced by the literatures in Chinese or others (PMID: 28508339, 20410600). 

6) Figure 1. “Arrows indicate…”. Only one arrow can be found in the panel B.

According to this suggestion, The “Arrows indicate …” has been corrected as “Arrow indicates …” in Fig. 1B of the revised version.

Reviewer 3 Report

This is an interesting article with a lot of information. I must say the article is very nicely written and the research was done well planned. I have some queries.

Major Comments:

In the abstract, in line 23, on page 1, the authors stated that sub-acute poisoning of 1,2- DCE up-regulates... It will be interesting to see the dose dependent response of 1,2-DCE at-least for some of the proteins. In line 166, on page 4, how did the authors precisely cut the equal hemispheres is not mentioned here. Its very important to know because of the water content measurement. It will be nice to see the NFKB activation by immunofluorescence imaging because by that one can show that the NFKB activation is localized to the site of edema formation. If the authors claim that all the genes are upregulated by NFkB signaling after 1,2 -DCE treatment, they should prove it with more confirmatory experiment like ChIP PCR at least  once. In the Fig. 1B of group b, the cells look more rounded in shape and the authors correctly noted that the perinuclear spaces had been increased after the 1,2 -DCE treatment. I am concerned whether the cells were undergoing apoptosis. Did the authors check whether the cells were undergoing apoptosis or not, maybe by TUNEL assay? In Fig. 4C and D the authors showed that in the control group the immunoreactivity of Iba 1, GFAP or VCAM1 is almost negligible, which is not the case, GFAP and Iba 1 as they are always expressed in astrocytes and microglia respectively. Using the enhancement of expression of GFAP and Iba 1 for the measurement of the activation of astrocytes or microglia might not reflect the scenario, and this is a topic of debate. However, the easier way to show the activation of microglia is its morphological changes after the treatment. Did the authors try to see the morphological changes of microglia at the site of activation? It will be nice to see a working model of the signaling pathways reported by the authors.

Minor Comments:

I believe there is a typo in line 19 of page 1. The authors are requested to correct it. In line 151, on page 4, the sentence doesn’t make sense to me. The authors are requested to please check it out. In line 254, on page 6, the authors might include the reference of Yun-Feng Li et al.2012, reproductive biology as they have already shown that NFKB activates MMP. What are the scale bars in the insets of Fig. 3D?

Author Response

This is an interesting article with a lot of information. I must say the article is very nicely written and the research was done well planned. I have some queries.

Major Comments:

1) In the abstract, in line 23, on page 1, the authors stated that sub-acute poisoning of 1,2- DCE up-regulates... It will be interesting to see the dose dependent response of 1,2-DCE at-least for some of the proteins.

In the previous studies (PMID: 25926354, PMID: 23263856), we have demonstrated the correlation of pathological changes in the liver or brain with the exposure doses of 1,2-DCE in mice. However, the brain edema occurred in mice only by treatment with the proper dosage of 1,2-DCE (PMID: 24964198). Otherwise, either no brain edema occurred or most mice were died. Therefore, it is difficult to observe the relationship of pathological changes in the brain with the exposure doses during the course of brain edema. However, the relationship of pathological changes in the brain with the exposure days has been reported in our previous study (PMID: 30081051). In the present study, since the contribution of neuroinflammatory reactions in the brain of 1,2-DCE-intoxicated mice to brain edema formation was mainly concerned, the pathological changes in the brain of mice exposed 1,2-DCE for three days were observed only.

2) In line 166, on page 4, how did the authors precisely cut the equal hemispheres is not mentioned here. Its very important to know because of the water content measurement.

The cerebral hemisphere was equally cut along sagittal sutures, which has been added in the revised version.

3) It will be nice to see the NFKB activation by immunofluorescence imaging because by that one can show that the NFKB activation is localized to the site of edema formation.

Activation of NF-KB has been checked by immunofluorescence staining in our previous in vitro experiments (PMID: 30087244), in which the nuclear translocation of p65 were increased in 2-CE treated astrocytes. According to this suggestion, we will try to do this in our next experiment though the immunoreactivity in tissue sections is not as good as cultured cells.

4) If the authors claim that all the genes are upregulated by NFkB signaling after 1,2-DCE treatment, they should prove it with more confirmatory experiment like ChIP PCR at least once.

In this study, we found that the increase in gene expression of MMP-9, ICAM-1, VCAM-1 and iNOS could be attenuated by pretreatment of the specific inhibitor to NF-KB. Gene expression was evaluated by the levels of target mRNA using real-time RT-PCR analysis that is the most common method used for this purpose at present time. However, we are also interested in the alternate method for determining gene expression levels. According to this suggestion, we will try to do this in our next experiment.

5) In the Fig. 1B of group b, the cells look more rounded in shape and the authors correctly noted that the perinuclear spaces had been increased after the 1,2 -DCE treatment. I am concerned whether the cells were undergoing apoptosis. Did the authors check whether the cells were undergoing apoptosis or not, maybe by TUNEL assay?

The pathological characteristics of brain edema observed under the light microscopy revealed lightly stained cytoplasm, swelling cell body (rounded in shape), enlarged perinuclear spaces and widened lacunar spaces surrounding vessels, which occurred due to accumulation of water in the intracellular or extracellular space. Apoptosis could be occurred following brain edema, however this subject was not included in this paper. According to this suggestion, apoptosis in the brain of 1,2-DCE-intoxicated mice will be considered in our further study.

6) In Fig. 4C and D the authors showed that in the control group the immunoreactivity of Iba 1, GFAP or VCAM1 is almost negligible, which is not the case, GFAP and Iba 1 as they are always expressed in astrocytes and microglia respectively. Using the enhancement of expression of GFAP and Iba 1 for the measurement of the activation of astrocytes or microglia might not reflect the scenario, and this is a topic of debate. However, the easier way to show the activation of microglia is its morphological changes after the treatment. Did the authors try to see the morphological changes of microglia at the site of activation?

Indeed, as shown in Fig. 4A, both GFAP and Iba-1 proteins were expressed detectable in the brain of control mice. Moreover, the expression levels of GFAP or Iba-1 were enhanced respectively due to activation in astrocytes and microglia. If we revealed them clearly in the control group of Fig 4C and D, the images captured using the same filming conditions in the treatment groups would be too strong to be seen clearly. Furthermore, the morphological changes of activated microglia could be observed in Fig. 4C, which showed a hypertrophy and increased proliferation of microglia in the treatment groups.

7) It will be nice to see a working model of the signaling pathways reported by the authors.

According to this suggestion, a working model of the signaling pathways shown in Figure 9 has been added in the revised version.

Minor Comments:

1) I believe there is a typo in line 19 of page 1. The authors are requested to correct it.

According to this suggestion, the typo in line 19 has been corrected in the revised version.

2) In line 151, on page 4, the sentence doesn’t make sense to me. The authors are requested to please check it out.

According to this suggestion, this sentence has been corrected in the revised version.

3) In line 254, on page 6, the authors might include the reference of Yun-Feng Li et al. 2012, reproductive biology as they have already shown that NF-KB activates MMP.

According to this suggestion, the reference of Li et al., 2012 has been cited in the section of discussion in the revised version.

4) What are the scale bars in the insets of Fig. 3D?

The scale bar in Fig. 3D represents 50 μm, which has been highlighted in the legend of Fig. 3D in revised version.